# Challenges in implementing the WHO-recommended package of care for advanced HIV disease in resource-constrained settings: A mixed-methods study

Temesgen Leka Lerango [1]*, Semalgn Leka Lerango[2], Mesfin Abebe [3], Tsion Mulat Tebeje[1], Habtamu Endashaw Hareru[1], Daniel Sisay [1], Getachew Assefa Zenebe[1], Yohannes Addisu[1], Biruk Bogale[4]

**1** School of Public Health, College of Health Sciences and Medicine, Dilla University, Dilla, Ethiopia,
**2** School of Medicine, College of Health Sciences, Addis Ababa University, Addis Ababa, Ethiopia,
**3** Department of Midwifery, College of Health Sciences and Medicine, Dilla University, Dilla, Ethiopia,
**4** JSI, Addis Ababa, Ethiopia

* maopharm@gmail.com

## Abstract

### Background

People diagnosed with advanced HIV disease (AHD) should be provided with the World Health Organization's (WHO) package of care to address their specific health-care needs. Although the WHO-recommended package of care is considered feasible and effective, its implementation remains sub-optimal across many sub-Saharan African (SSA) countries. This study aimed to explore challenges in implementing the WHO-recommended package of care for advanced HIV disease in resource-constrained settings.

### Methods

A sequential explanatory mixed-methods study was conducted between March 1 and April 30, 2024, in the Gedeo Zone of Southern Ethiopia. The quantitative data involved extraction from medical records of 145 individuals newly diagnosed with AHD. For the qualitative inquiry, healthcare providers engaged in the HIV care continuum were purposively selected for in-depth key informant interviews. An inductive thematic analysis was conducted to identify and interpret recurrent patterns within the qualitative data. Quantitative data were analyzed using R version 4.3.3, while qualitative data were organized and managed using NVivo version 14.

### Results

Only about half (47.6%) of the newly diagnosed AHD cases underwent baseline CD4 count testing. All 145 individuals were screened for TB using the WHO four-symptom

**Data availability statement:** All relevant data is included in the manuscript and the Supporting Information file. Data collected as part of qualitative research is provided as a codebook in Supporting Information file. Raw data cannot be shared publicly because data contain potentially identifying or sensitive patient and participant information. Data are available from the corresponding author (maopharm@gmail.com or temesgenleka@du.edu.et) upon reasonable request for researchers who meet the criteria for access to confidential data and subject to ethics committee approval. Researchers interested in accessing the raw data may communicate IRB of College of Medical and Health Sciences at Dilla University (contact via email to: duirbsubmission@gmail.com). Researchers can also reach out to the corresponding author for assistance.

**Funding:** The author(s) received no specific funding for this work.

**Competing interests:** The authors have declared that no competing interests exist.

**Abbreviations**: AHD, Advanced HIV disease; ART, Antiretroviral therapy; APTS, Auditable Pharmaceutical Transactions and Services; CD4, Clusters of differentiation 4; CrAg, Cryptococcal antigen; DSD, Differentiated service delivery; EPSA, Ethiopian Pharmaceutical Supply Agency; FMoH, the Federal Ministry of Health, Ethiopia; HIV, Human immunodeficiency virus; KII, Key informant interview; LF-LAM, Lateral flow urine lipoarabinomannan assay; MTB, *Mycobacterium tuberculosis*; RIF, Rifampin; OIs, Opportunistic infections; SDI, Same-day ART initiation; TB, Tuberculosis; UNAIDS, The Joint United Nations Programme on HIV/AIDS; WHO, World Health Organization.

algorithm, and 78.6% underwent confirmatory GeneXpert® MTB/RIF testing. Among individuals with AHD, 92.4% received co-trimoxazole prophylaxis, and 14.5% received tuberculosis preventive therapy. Rapid ART initiation was implemented for 20.0% of individuals with AHD. All newly diagnosed individuals with AHD received tailored counseling to ensure optimal adherence. Qualitative data analysis identified three principal challenges to the implementation of the WHO-recommended package of care: structural and organizational obstacles, service delivery constraints, and patient-related concerns as expressed by healthcare workers.

## Conclusions

The implementation of the WHO-recommended package of care for individuals with AHD remains inconsistent. Although adherence support is routinely offered to all newly diagnosed individuals with AHD, the delivery of other key components is frequently hindered by a range of systemic challenges. These include the unavailability or frequent stockouts of essential medications and services for managing opportunistic infections, weak referral and linkage systems, and the absence of dedicated AHD care clinics. Such challenges underscore significant gaps in the continuum of AHD care and highlight the pressing need for targeted, system-level interventions to ensure comprehensive service delivery.

## Introduction

Human Immunodeficiency Virus (HIV) continues to be one of a major causes of morbidity and mortality in many countries, especially in sub-Saharan Africa (SSA) [1]. At the end of 2024, approximately 40.8 million people worldwide were living with HIV, including 1.3 million new infections, of whom 31.6 million were receiving antiretroviral therapy (ART). Globally, an estimated 630,000 HIV-related deaths occurred in 2024, of which approximately 60.3% were in the African Region [2]. Projections indicate that sustaining the current levels of HIV control is insufficient to meet the ambitious targets for reducing incidence and mortality by 2030 [3].

Despite the widespread roll-out of ART, many individuals in SSA still cannot access treatment until the later stages of their illness, often presenting with advanced HIV disease (AHD) [4,5]. A CD4 count below 200 cells/mm$^3$ or WHO clinical stage 3 or 4 defines AHD in adults, adolescents, and children aged five and older. All children under five with HIV are considered to have AHD [6,7]. The global prevalence of AHD in healthcare settings, including both inpatient and outpatient care, is estimated at 33.7%, with no evidence of significant change in recent years [8]. From 2016 to 2019, the prevalence of AHD among individuals initiating ART with a baseline CD4 count ranged from 32% to 40% across the four African regions, excluding South Africa [9]. In SSA, an estimated 1.88 million people are living with AHD [10].

Individuals with AHD are more susceptible to opportunistic infections (OIs) such as tuberculosis (TB), severe bacterial infections, and cryptococcal meningitis (CM)

[11,12]. Moreover, their risk of mortality increases as CD4 cell counts decline, and this risk has remained stable in recent years [13,14].

Using a Differentiated Service Delivery (DSD) approach, the WHO's recommended intervention package for AHD, which includes screening and diagnosis, early ART initiation, prevention and management of major OIs, and adherence support, should be provided to all individuals presenting with AHD [12,15]. National HIV programs are encouraged to adopt the WHO-recommended package, which will be updated as new evidence and insights emerge [1]. These strategies would help reduce HIV incidence and HIV-related mortality [16].

Despite the high burden of OIs, diagnosing and treating them remains challenging and limited in resource-constrained settings [17]. International funding has been crucial in reducing new HIV infections and related deaths in SSA. Funding cuts could reverse this progress by 2030, disproportionately impacting SSA and vulnerable populations [18–23]. On the other hand, the defunding of conventional lab-based CD4 testing has left treatment programs dependent on a few rapid tests with variable performance, further compounded by the withdrawal of key manufacturers of CD4 cell count tests from the market in low- and middle-income countries [17].

Advanced HIV poses a significant challenge and remains an underexplored area of HIV care [24]. Over the last decade, advanced HIV has become neglected, with insufficient resources and inadequate research focused on its prevention, screening, diagnosis, and treatment [17]. Effective implementation of the AHD care package has the potential to improve the diagnosis of OIs and enhance the overall quality of care for individuals with AHD [25]. Although the implementation of the WHO-recommended package of care is feasible, it has been undertaken to only a limited extent in many sub-Saharan African countries [26].

Understanding the challenges in implementing WHO-recommended package of care for AHD patients in resource-constrained settings can inform policy and programmatic changes to improve care for patients with AHD. Therefore, our study aimed to explore challenges in implementing the WHO-recommended package of care for advanced HIV disease in resource-constrained settings. The findings will help inform strategies to improve the implementation of the WHO-recommended care packages for AHD and support decision-makers in HIV programs to allocate resources more effectively.

## Materials and methods

### Study setting

Gedeo Zone is part of the South Ethiopia Regional State (SERS), Ethiopia. It is surrounded by the Oromia Region, which borders the zone on the east, south, and west. Gedeo shares its northern boundary with the Sidama Region. Dilla, located on the main road from Addis Ababa to Nairobi, is the administrative center. The 2007 Census of Ethiopia reports that this zone has a total population of 847,434, of whom 424,742 are men and 422,692 are women.

HIV care services are provided at eight public health facilities in the Gedeo Zone, Southern Ethiopia. These include Dilla University General Hospital, Bule Primary Hospital, Gedeb Primary Hospital, Yirgachefe Primary Hospital, Dilla Health Center, Wonago Health Center, Gerse Health Center, and Chelelktu Health Center. Dilla University General Hospital serves as the hub center for HIV service delivery in the Gedeo Zone, while the other facilities operate as spoke centers.

### Study design and period

A sequential explanatory mixed-methods study was conducted between March 1 and April 30, 2024, in the Gedeo Zone of Southern Ethiopia. A mixed-methods approach was deemed appropriate for our research due to the inclusion of two sequential phases of data collection [27,28].

### Population

For the quantitative component of the study, relevant data were extracted from the medical records of individuals newly diagnosed with advanced HIV and initiating antiretroviral therapy at five selected healthcare facilities: Dilla University

General Hospital, Yirgachefe Primary Hospital, Gedeb Primary Hospital, Dilla Health Center, and Wonago Health Center. Of the 422 newly diagnosed people living with HIV who initiated antiretroviral therapy, 145 were identified as having advanced HIV disease. The quantitative analysis in this study was based on the medical records of these 145 individuals, collected between May 29, 2023, and February 6, 2024.

Meanwhile, the qualitative component of the study explored the in-depth perspectives of healthcare providers involved in HIV care, drawing on their professional expertise and experiences across the HIV care continuum. Heterogeneous purposive sampling was employed to select providers for in-depth key informant interviews, with the aim of capturing a broad range of informed viewpoints on the challenges faced during the implementation of the WHO-recommended package of care for patients with AHD. In-depth interviews were conducted with eight key informants out of the ten initially planned, as data saturation had been achieved.

### Definition of terms

**Advanced HIV disease**: World Health Organization defines advanced HIV disease for adults and adolescents (and children five years and older) as having a CD4 cell count of less than 200 cells/mm$^3$ or WHO clinical stage 3 or 4 disease. All children younger than five years living with HIV are considered to have advanced HIV disease [6,7].

**WHO recommendation package**: A package of interventions including screening, treatment and/or prophylaxis for major opportunistic infections, rapid ART initiation, and intensified adherence support should be offered to everyone presenting with advanced HIV disease [29].

**Rapid ART initiation**: ART initiation within seven days from the day of HIV diagnosis, calculated as the number of days between HIV diagnosis and ART enrollment [12].

### Data collection procedures

Sequential explanatory design was employed for data collection. A data abstraction checklist for quantitative data and an interview guide for qualitative data were developed based on an extensive review of the WHO's *Consolidated Guidelines on HIV Prevention, Testing, Treatment, Service Delivery, and Monitoring: Recommendations for a Public Health Approach* [7], ensuring the inclusion of relevant information.

Initially, quantitative data were extracted from clinical records using a standardized data abstraction checklist. Five clinical nurses from ART clinics carried out the data extraction process under the supervision of two public health officers. The data enumerators used Android devices with the Kobo Collect app installed. They downloaded blank forms from the Kobo Toolbox server and filled them out to extract relevant data. Data collectors and supervisors underwent rigorous training on the study objectives and data collection procedures. The principal investigator systematically reviewed the submitted datasets to ensure their accuracy and completeness.

To ensure the effectiveness of qualitative data collection, the interview guide was pre-tested through simulation interviews with data collectors and healthcare providers who were not part of the selected study participants. The guide was revised in response to the findings. In-depth key informant interviews (KIIs) were conducted in Amharic, with each interview lasting 15–30 minutes. The sessions were audio-recorded, transcribed verbatim, and subsequently translated into English. The translated transcripts were then returned to participants for review and confirmation of accuracy.

### Data processing and analysis

**Quantitative data.** Data were downloaded from the Kobo Toolbox web server as excel files. Quantitative analysis was performed using R version 4.3.3 (R Core Team, Vienna, Austria) [30]. Descriptive statistics were used to summarize background characteristics of PLHIV with AHD. Participants' continuous characteristics were summarized as mean with standard deviation or median with interquartile range, while categorical variables were presented as frequencies with their respective percentages.

## Qualitative data

TLL, TMT, and BB conducted independent transcript reviews to verify accuracy. Following thorough familiarization with the data, a codebook (S1 Table) was developed using the qualitative data management software NVivo 14 for Windows. An inductive thematic analysis was performed to identify patterns within the interview data. Subsequently, the data were abstracted and analyzed, with illustrative direct quotations from participants integrated into the final report to support and enrich the findings.

## Ethics statement

This study was reviewed and approved by the Institutional Review Board of the College of Medical and Health Sciences at Dilla University with the approval number: duirb/035/23–05, dated May 18, 2023. All procedures adhered to the ethical guidelines outlined in the Declaration of Helsinki for human research. All participants provided written informed consent to participate in the study. The data were collected anonymously, kept confidential, and secured throughout the study.

## Results

### Background characteristics of participants

**People living with AHD.** The mean age of the study participants was 33.5 years (SD ± 8.9), with more than half (55.9%) aged between 30 and 49 years. Two-thirds (66.2%) of the participants lived in rural areas. Nearly half (48.9%) had completed primary education, and just over half (53.1%) were married. Additionally, 56.6% of the participants were classified as underweight. The majority of participants (84.1%) were classified as WHO clinical stage 3. Of the 145 AHD cases, only 69 (47.6%) had a recorded baseline CD4 count. Among these 69 cases, two-thirds (n = 46, 66.7%) had a CD4 count between 100 and 199 cells/mm$^3$ (Table 1).

### Key informant participants

The median (IQR) age of KIs was 36.5 years (34.75–42). The majority were male (7, 87.5%). Half of the KIs were public health officers (4, 50%), while the others included two clinical nurses, one laboratory technologist, and one pharmacist. Most held a bachelor's degree (6, 75%). The median (IQR) duration of HIV care experience was 3.5 years (2.75–4) (Table 2).

### Adherence to the WHO-recommended package of care

The components of the care package for individuals with advanced HIV disease are listed in S2 Table, and the algorithm for delivering the care package is presented in S1 Fig [7].

### Screening and diagnosis

A WHO-recommended four-symptom TB screening was performed in all 145 AHD cases. Of those screened, 127 (87.6%) tested positive, and 18 (12.4%) tested negative.

GeneXpert® MTB/RIF testing, used as the initial diagnostic tool for TB, was performed on 114 individuals with AHD. Among those tested, 82 (71.9%) were positive for TB. A urine lateral flow urine lipoarabinomannan assay (LF-LAM) test was conducted on 92, with 78 (84.8%) testing positive. None of the individuals with AHD in these settings had been tested for cryptococcal antigen (CrAg) from May 29, 2023, to February 06, 2024.

### Prophylaxis and pre-emptive treatment

Of the participants with AHD, 134 (92.4%) received co-trimoxazole prophylaxis, 21 received TB preventive treatment, and 33 received fluconazole primary prophylaxis.

**Table 1. Background characteristics of newly diagnosed people living with AHD initiating ART in the Gedeo Zone, Southern Ethiopia (n = 145).**

| Variables | Frequency (n) | Percent (%) |
|---|---|---|
| **Age (in years)** | | |
| 18–29 | 56 | 38.6 |
| 30–49 | 81 | 55.9 |
| ≥ 50 | 8 | 5.5 |
| **Sex** | | |
| Female | 73 | 50.3 |
| Male | 72 | 49.7 |
| **Place of residence** | | |
| Rural | 96 | 66.2 |
| Urban | 49 | 33.8 |
| **Educational status** | | |
| No formal education | 53 | 36.6 |
| Primary education | 71 | 48.9 |
| Secondary and above education | 21 | 14.5 |
| **Employment status** | | |
| Housewife | 42 | 29.0 |
| Private work | 40 | 27.6 |
| Public servant | 7 | 4.8 |
| Unemployed | 19 | 13.1 |
| Others˙ | 37 | 25.5 |
| **Marital status** | | |
| Single | 17 | 11.7 |
| Married | 77 | 53.1 |
| Divorced | 22 | 15.2 |
| Widowed | 29 | 20.0 |
| **Average monthly income (in ETB)** | | |
| < 400 | 45 | 31.0 |
| 400–3000 | 74 | 51.0 |
| > 3000 | 26 | 18.0 |
| **Body mass index (BMI) (in kg/m²)** | | |
| Underweight (BMI < 18.5) | 82 | 56.6 |
| Normal (18.5–24.9) | 62 | 42.7 |
| **WHO clinical staging** | | |
| Stage 1 | 1 | 0.7 |
| Stage 2 | 9 | 6.2 |
| Stage 3 | 122 | 84.1 |
| Stage 4 | 13 | 9.0 |
| **CD4 count* (cells/mm³)** | | |
| < 100 | 23 | 33.3 |
| 100–199 | 46 | 66.7 |

Others˙ – daily laborers, * Baseline CD4 count test was performed for only 69 PLHIV.

**Table 2. Background characteristics of key informant participants.**

| Variables | Frequency | Percent (%) |
|---|---|---|
| **Sex** | | |
| Male | 7 | 87.5 |
| Female | 1 | 12.5 |
| **Profession** | | |
| Public Health Officer (PHO) | 4 | 50.0 |
| Clinical Nurse (CN) | 2 | 25.0 |
| Medical Laboratory Technologist (MLT) | 1 | 12.5 |
| Pharmacist | 1 | 12.5 |
| **Levels of education** | | |
| Masters | 2 | 25.0 |
| Bachelor | 6 | 75.0 |
| **Age (in years)**: median (IQR) = 36.5 (34.75–42). | | |
| **Professional experience (in years)**: median (IQR) = 11 (9.75–17.75). | | |
| **Experience in HIV care (in years)**: median (IQR) = 3.5 (2.75–4). | | |

## Antiretroviral therapy initiation

The median (IQR) duration from diagnosis to ART initiation was 15 days (14–18). The majority of participants (n = 132, 91%) started the TDF + 3TC + DTG regimen. Four individuals (2.8%) have initiated ART on the same day of their diagnosis. Rapid ART initiation was carried out in twenty-nine (20.0%) of the 145 individuals with AHD.

## Adapted adherence support

All newly diagnosed individuals with advanced HIV initiating antiretroviral therapy in the current settings received tailored counseling to ensure optimal adherence.

## Qualitative findings

This study aimed to explore the challenges in implementing the WHO's package of care for newly diagnosed individuals with AHD in resource-constrained settings. An inductive thematic analysis was employed to identify patterns across interviews. Three principal challenges emerged from qualitative data analysis: structural and organizational obstacles, service delivery constraints, and patient-related concerns as expressed by healthcare workers.

## Theme 1: Structural and organizational obstacles

This theme encompasses critical structural and organizational limitations that affect the effective provision of care packages for AHD patients, such as lack of specialized AHD clinics, supply chain constraints, service interruptions, and a non-comprehensive AHD registry.

**Subtheme 1.1: Lack of specialized AHD clinics.** Healthcare providers consistently underscore the need for differentiated service delivery for individuals with AHD. Given their heightened risk of morbidity and mortality, and presentation with advanced clinical conditions, these patients require more specialized and targeted care interventions. Healthcare providers have expressed concern about the lack of specialized AHD clinics as follows:

"*When AHD services are delivered alongside other ART services, AHD patients often do not receive the full level of care they need. Under heavy workload, caregivers may treat AHD patients the same as others, despite their need*

for more focused care. For example, key populations (KP) receive differentiated services, and implementing a similar approach for AHD patients could lead to improved outcome." (P4, PHO)

Additionally, the lack of dedicated specialized AHD clinics poses challenges in obtaining adequate consultation from senior physicians.

**Subtheme 1.2: Supply chain constraints.** In resource-constrained settings, supply chain and logistical constraints significantly affect the availability of essential medications and health services. These challenges have multifaceted causes, including inadequate funding, limited local manufacturing capacity, transportation-related difficulties, and insufficient knowledge of health commodity supply management.

In Ethiopia, medical resources required for HIV care are provided through a program-based approach, supported by the Ethiopian Federal Ministry of Health (FMoH) and other stakeholders involved in the HIV program. In recent years, the supply of health commodities for HIV services, particularly OI medications, has significantly declined. The problem lies not only in reduced stock levels but also in inconsistent supply. The pharmacist expresses his concern as follows:

*"Over the past two years, the supply of OI medications has been inadequate. Although we report the Report and Requisition Format (RRF) to EPSA, the supply of OI medications is currently inadequate and frequently interrupted."* (P6, Pharmacist)

Health facilities receive health commodities for HIV care through a top-down supply system. The primary problem with the top-down (push) supply system is that it often results in a mismatch between demand and supply, which in most cases leads to stockouts. Supply chain constraints have a detrimental effect on the availability of OI medications and nutritional supplements.

Many patients present with OIs and require appropriate medications for their management. To improve their quality of life, eligible patients should receive OI management alongside ART. However, they currently face persistent challenges in accessing OI medications through ART pharmacies.

On the other hand, the shortage of nutritional supplements poses an additional challenge to delivering care in accordance with established guidelines. Care providers emphasize the difficulties they face in managing malnourished patients because of this shortage.

*"Although there are guidelines to manage malnourishment based on BMI, we lack the means to address it effectively. This remains a major challenge, as malnourishment can lead to other OIs like TB, and many patients are too weak to tolerate antiretroviral therapy."* (P2, Senior PHO)

Furthermore, malnourished patients are often reluctant to take ART medications and frequently request nutritional support.

*"With some nutritional support, at least they can continue taking their ART medications. They often ask, 'What are we going to eat in order to take these medications?'"* (P1, CN)

When patients are unable to access necessary medications, they may lose confidence in the care provided and disengage from follow-up services. Additionally, insufficient nutritional supplementation poses challenges not only in managing malnutrition but also in supporting the proper intake of ART medications.

**Subtheme 1.3: Service interruptions due to resource shortages.** Resource shortages lead to interruptions in certain services. Participants report experiencing disruptions in LF-LAM and GeneXpert® services. Healthcare providers describe the service interruptions they encountered during service delivery as follows:

*"There have been some interruptions in the availability of LF-LAM, but it remains a valuable diagnostic tool."* (P2, Senior PHO)

*"To some extent, there have been occasional interruptions in the GeneXpert® testing service due to cartridge shortages."* (P4, MLT)

In addition to service inadequacies, interruptions in service provision pose a significant challenge to the delivery of adequate care.

**Subtheme 1.4: Auditable Pharmaceutical Transactions and Services (APTS)-related challenges.**  The APTS system contributes to ensuring the availability of and access to essential medicines in facilities where it is implemented. While it has improved standard practices in recording and reporting, and enhanced transparency in the management of pharmaceutical commodities, it has also restricted the practice of drug exchange among different pharmacy units within health facilities.

*"We used to engage in drug exchange practices with dispensaries in other units, but the new APTS system has restricted such practices. Most of our patients are in desperate situations and have low socio-economic status. Seeing the burden they endure due to OIs, we tried to explore the possibility of obtaining OI medications through drug exchange. However, the APTS system does not permit such practices."* (P2, Senior PHO)

The restriction of drug exchange practices under the APTS system is one of the factors contributing to the inability of AHD patients to access OI medications free of charge.

**Subtheme 1.5: Non-comprehensive AHD registry.**  Another challenge clinicians face in AHD care is the non-comprehensiveness of the current AHD registry format. Patients with AHD are typically followed on a monthly basis until their condition stabilizes, after which longer appointment intervals are scheduled. Accordingly, clinicians are expected to monitor patients' progress through regular assessments at each visit, in line with national guidelines. One clinician describes the difficulties encountered in documenting patients' follow-up progress as follows:

*"Regarding the AHD registry format, it is not comprehensive. For example, patients with TB require follow-up for six months, but the follow-up section in the registry only lists the 1st, 2nd, and 3rd visits without specifying the intervals between them. The registry does not capture the activities performed during each visit; instead, these are documented separately in the patient's chart."* (P2, Senior PHO)

The non-comprehensiveness of the registry limits caregivers' ability to document patients' prognostic information in detail, thereby hindering the provision of quality care, particularly for those requiring extended follow-up.

## Theme 2: Service delivery constraints

This theme highlights challenges in service delivery, including ineffective referral and linkage systems, limited availability of services, and the underutilization of available services by spoke facilities, as well as workforce-related issues to the implementation of the AHD care package.

**Subtheme 2.1: Ineffective referral and linkage.**  The FMoH recommends referring AHD patients from spoke facilities to hub centers for advanced diagnosis and management, and returning them to spoke sites for follow-up after clinical stabilization. However, despite clear referral guidelines, implementation remains suboptimal.

*"…, the referral system is not well-structured and lacks consistency. An effective referral system is essential to ensure that patients receive appropriate care. Directing patients to facilities (Hub) where they can access the necessary services is a crucial component of effective disease management."* (P2, Senior PHO)

This suboptimal referral practice at spoke facilities hinders the timely provision of essential services, such as advanced TB diagnostics and the management of complex conditions like cryptococcal meningitis.

**Subtheme 2.2: Inadequate service availability.** Some essential services, such as baseline testing, are not fully available in all HIV care–providing facilities. For instance, the CrAg assay was only recently introduced at a hub center that serves spoke facilities within its catchment area.

*"It has been nearly a year since baseline CrAg testing was introduced at our facility. Before the test was introduced, patients who had already started ART did not undergo baseline CrAg testing."* (P3, MLT)

This inadequacy in service provision hinders the timely and appropriate diagnosis and management of patients, thereby contributing to an increased burden of morbidity and mortality among individuals with AHD.

**Subtheme 2.3: Underutilization of services by spoke facilities.** Another concerning issue is the underutilization of available services by the spoke facilities. In Ethiopia, hub centers are designated to provide essential services to spoke facilities within their catchment areas. However, despite clear directives from the FMoH, the utilization of these services remains suboptimal. One service provider at a hub center describes this challenge as follows:

*"Our facility provides baseline CD4 testing services to facilities (spoke) within our catchment area. Although there are some challenges with utilization by these facilities, we address them through mentoring, as baseline CD4 testing is essential for effective care provision."* (P2, Senior PHO)

Due to underutilization, many newly diagnosed individuals enrolling in ART at spoke facilities are not receiving essential baseline investigations such as CD4 count testing and the CrAg assay.

**Subtheme 2.4: High workload and inadequate human power.** Some health facilities lack sufficient staff to manage the workload. The heavy patient load occasionally compromises the quality of care. A clinician working in such an environment expresses his concerns as follows:

*"At our hospital, there is a significant workload. Numerous registry formats must be completed, and several patient investigations, such as CD4 testing, viral load monitoring, and cervical cancer screening, are required. However, time constraints negatively impact the quality of care provided by health professionals. For example, when I receive 10 to 15 patient charts at once, I may overlook some of the necessary investigations."* (P4, PHO)

With a high workload, healthcare providers may not have adequate time for thorough patient assessments and may overlook recommended protocols, leading to suboptimal care.

**Subtheme 2.5: Insufficient training.** Few staff members have received training on the AHD package of care, and even that is insufficient. Others have not received any formal training, relying solely on reading the guidelines. Staff who depend on guidelines and learning from colleagues underscore the necessity of training as follows:

*"Regarding training, even I at the zonal level have not received training on AHD. During supervision and mentorship, we provide feedback on practice gaps at service delivery points based on our knowledge of the relevant guidelines.."* (P7, PHO at ZHB)

Insufficient training may result in misdiagnosis, improper treatment, and compromised quality of care.

## Theme 3: Patient-related concerns as expressed by healthcare workers

Healthcare providers report that certain patient-related factors, including clinical conditions, financial constraints, and psychological unreadiness, hinder the delivery of guideline-recommended care.

**Subtheme 3.1: Clinical factors.**  Most patients with AHD present with advanced disease and OIs, requiring clinical stabilization before initiating ART. As a result, same-day or rapid ART initiation is often delayed.

*"Patients in WHO clinical stages 1 and 2 start ART immediately after giving their consent, while those in stage 3 or 4 face a higher pill burden for managing their condition, so ART initiation is delayed for a short period."* (P2, Senior PHO)

**Subtheme 3.2: Psychological unreadiness.**  At times, beyond clinical and service-related factors, ART initiation is delayed due to patients' psychological unreadiness to accept their diagnosis. Some patients are adamant in rejecting the test results, and their enrollment into ART is postponed until they come to terms with the diagnosis and are psychologically prepared to begin treatment. A public health officer described such a scenario as follows:

*"Some patients do not accept their test results and request additional testing at other facilities to confirm the diagnosis. In such cases, we allow them for a period of up to one week or more to make an informed decision regarding enrollment in HIV care."* (P5, PHO)

**Subtheme 3.3: Financial constraints.**  Financial constraints pose a major challenge to the implementation of care packages. Although many patients require OI management, medications are often unavailable in ART pharmacy, forcing patients to purchase them from private pharmacies. Given the severe socioeconomic disadvantage of most AHD patients, many remain untreated, leading to disease progression.

*"AHD patients do not purchase medicines from retail drug outlets because they cannot afford them. When they are unable to take their medications, their condition worsens, they get sick, stay at home, and often do not return for follow-up."* (P1, CN)

The three principal challenges in implementing the WHO-recommended package of care for AHD in our study settings are summarized in Table 3.

## Discussion

The primary goal of delivering the WHO's package of care interventions for individuals with advanced HIV disease is to prevent, diagnose, and treat the most common causes of HIV-related morbidity and mortality. Although the WHO's intervention package for AHD is based on evidence-based recommendations, implementation of the package of care was sub-optimal in our study settings. Likewise, a study evaluating the screening, diagnosis, and treatment of AHD in an urban setting in Southwestern Uganda demonstrated sub-optimal implementation of the AHD intervention package [31]. Similarly, a study conducted in a rural setting in Malawi evaluating adherence to WHO's AHD management guidelines found sub-optimal adherence to these guidelines among patients with AHD [32]. Sub-optimal implementation of established recommendations, compounded by recent funding cuts, could reverse global efforts to end the AIDS epidemic as a public health threat by 2030 [18,23].

Individuals with AHD who are eligible for a package of care can be identified using CD4 cell count, which is the best indicator of disease stage and risk of death. Although WHO clinical staging can be used to identify eligible individuals, relying on it alone risks missing a substantial number of people living with HIV who have severe immune suppression. In addition to identifying individuals eligible for the recommended care package, CD4 count testing has the potential to avert deaths from TB and CM, which are two of the most deadly OIs among patients with AHD [33]. Therefore, baseline CD4 count testing remains important for all PLHIV enrolling in ART [7,34].

Despite its critical importance, baseline CD4 cell counts were performed for only 47.6% of advanced HIV cases in our study settings. This implies that more than half of the cases were identified using the WHO clinical staging. Consequently,

**Table 3. Summary of challenges in implementing the WHO-recommended package of care for advanced HIV disease.**

| Themes | Subthemes | Description |
|---|---|---|
| **Structural and organizational obstacles** | Lack of specialized AHD clinics | AHD patients do not receive adequate care due to the absence of differentiated AHD services. |
| | Supply chain constraints | Supply chain and logistical constraints limit the availability of essential medications and services. |
| | Service interruptions | Resource shortages have caused interruptions in essential services, such as baseline CD4 testing and LF-LAM. |
| | APTS-related challenges | The restriction of within-facility drug exchange practices under the APTS system hinders access to OI medications. |
| | Non-comprehensive AHD registry | The limited scope of the AHD registry impedes detailed documentation of prognostic information. |
| **Service delivery constraints** | Ineffective referral and linkage | The referral system is poorly structured and inconsistent, hindering the timely provision of essential services. |
| | Inadequate service availability | Limited availability of services, such as baseline testing, hampers accurate diagnosis and effective management. |
| | Underutilization of services by spoke sites | Essential services available at the hub center are underutilized by spoke facilities. |
| | High workload | High workload from routine tasks and a large patient load in the ART clinic negatively impacts quality of care. |
| | Insufficient training | Insufficient professional training on AHD leads to misdiagnosis, improper treatment, and reduced quality of care. |
| **Patient-related concerns as expressed by healthcare workers** | Clinical factors | Most AHD patients require clinical stabilization before ART initiation, causing delays. |
| | Psychological unreadiness | Some patients adamantly reject the test results, delaying their enrollment in ART. |
| | Financial constraints | AHD patients with low socio-economic status cannot afford to buy medicines from retail drug outlets and many remain untreated. |

a considerable proportion of individuals with advanced HIV might have gone unidentified, potentially missing the package of care for which they were eligible. Robust evidence from the two meta-analyses assessing the diagnostic accuracy of the WHO clinical staging system for detecting AHD indicated that a significant proportion of PLHIV may be misclassified, leading to missed opportunities for the prophylaxis, diagnosis, and treatment of opportunistic infections [35,36]. This finding underscores the critical need to enhance CD4 test availability at service delivery points, enabling a comprehensive evaluation of PLHIV enrolling in care. Adequate attention should be given to ensuring the availability of baseline CD4 testing services at spoke facilities. This is supported by evidence from a study that explored global trends in CD4 count measurement at the initiation of ART, highlighting the need for widespread adoption and adequate funding for baseline CD4 measurement at ART initiation [9].

Point-of-care diagnostics theoretically have the potential to reduce mortality by addressing laboratory delays and providing timely same-day results to patients and healthcare workers [37]. Therefore, enhancing the availability of diagnostic resources could potentially improve health outcomes among PLHIV [32]. In spoke facilities lacking diagnostic capacity, mentoring and supporting healthcare professionals to strengthen referral and linkage of newly diagnosed AHD patients to hub centers for essential investigations and care is crucial in resource-constrained settings. Ongoing mentoring and performance monitoring are essential for effective AHD care implementation and improved outcomes [24,38,39]. A growing body of evidence supports that capacity-building training in AHD care can enhance outcomes across the entire care cascade [40].

Screening for OIs, such as TB and CM, is crucial due to the high mortality associated with these co-infections [41]. Strengthening screening and diagnostic efforts for AHD-related OIs is imperative to improve clinical outcomes and ensure progress in HIV care delivery [42]. Adults and adolescents living with HIV should be screened for TB using a clinical algorithm. Those who do not report symptoms such as a current cough, fever, weight loss, or night sweats are unlikely to have active TB and should be offered preventive treatment, regardless of their ART status. After ruling out active TB disease, twenty-one individuals were provided with TB preventive treatment in the current settings. In our study settings, the performance of the four-symptom TB screening aligned with WHO recommendations, ensuring that all individuals with AHD were screened for TB. GeneXpert® MTB/RIF testing was performed on 89.7% (114/127) of individuals who screened positive for TB, while 10.2% (13/127) were not tested. To assist in diagnosing TB, LF-LAM testing was performed among individuals with symptoms and signs of the disease, with 84.4% (78 out of 92) testing positive. There are occasional interruptions in TB diagnostics in the current settings due to resource shortages. To avoid such interruptions and ensure continuity in diagnostic services, well-planned and sustainable strategies should be implemented that guarantee consistent resource availability.

Globally, nearly a fifth (19%) of AIDS-related mortality is attributed to cryptococcal disease [43]. The WHO recommends that countries prioritize reliable access to rapid CrAg diagnostic assays, preferably lateral flow assays, for use in cerebrospinal fluid (CSF), serum, plasma, or whole blood [7]. Screening for cryptococcal infection in PLHIV can identify asymptomatic individuals, allowing for early treatment and prevention of mortality [44]. There is an urgent need to scale up cryptococcal diagnostics, access to effective meningitis treatment, and preventive screening to eliminate cryptococcal meningitis–related mortality by 2030 [43]. Cryptococcal antigen test is affordable, easy to use, and highly sensitive in CM detection [45]. In addition to strengthening diagnostic and treatment capacity for CM, it is necessary to raise awareness among healthcare workers about the impact of fungal infections in AHD and to provide technical assistance and additional training [46][47].

Antiretroviral therapy alone is insufficient for the effective management of CrAg-positive PLHIV; therefore, treatment programs in resource-constrained settings should integrate CrAg screening into HIV care, particularly targeting individuals with CD4 cell count below 100 cells/mm$^3$ [48,49]. Existing evidence supports routine screening for all PLHIV with CD4 cell counts below 100 cells/mm$^3$, and recommends considering screening for those with CD4 counts below 200 cells/mm$^3$ [50]. In our setting, none of the individuals with AHD had been tested for CrAg. This finding is consistent with a study on the management of AHD among individuals initiating ART in Senegal, which reported that no participants underwent CrAg screening or received fluconazole for preventive therapy [51]. A recent study assessing clinical guideline use in CM care among healthcare workers in Ethiopia demonstrated suboptimal uptake, with only 29% reporting routine implementation [46]. Similarly, a study assessing adherence to standards of care in the management of HIV patients at risk of CM in Uganda reported that only 19% of eligible patients underwent CrAg screening [52]. Even when no CrAg testing was performed, 33 individuals with AHD in our study settings received fluconazole primary prophylaxis in accordance with WHO recommendations [29]. This study highlights gaps in CrAg testing for individuals with AHD in resource-limited settings and underscores the urgent need to expand testing and ensure the availability of medications to prevent cryptococcal infections. Failure to screen for CrAg among individuals with AHD represents a critical missed opportunity for early intervention and improved clinical outcomes [53]. Antiretroviral therapy initiation should be deferred for four to six weeks after starting antifungal treatment in individuals with CM to minimize the risk of mortality [54].

Co-trimoxazole prophylaxis is recommended for adults (including pregnant women) with severe or advanced HIV clinical disease (WHO stage 3 or 4) or with CD4 cell count ≤350 cells/mm$^3$ [7]. In the current settings, 92.4% of people with AHD have received co-trimoxazole prophylaxis. Co-trimoxazole prophylaxis should be initiated regardless of CD4 cell count or WHO clinical stage in settings where malaria or severe bacterial infections are highly prevalent [55]. Malaria is highly prevalent in the Gedeo Zone, particularly in Dilla Town and its surrounding areas [56]. According to the recommendation, the implementation of co-trimoxazole prophylaxis in the study settings was satisfactory.

According to consolidated guidelines on the use of antiretroviral drugs for treating and preventing HIV infection, ART should be initiated for all people living with HIV regardless of WHO clinical stage and at any CD4 cell count [57]. Following confirmed HIV diagnosis and clinical assessment, PLHIV should initiate ART without delay. Same-day ART initiation (SDI) should be offered to those who are clinically eligible and ready to begin treatment. Same-day initiation was found to be effective in achieving viral suppression [58,59]. Likewise, rapid initiation of antiretroviral therapy was associated with a significant reduction in mortality [60]. Despite its benefits, same-day initiation was associated with lower retention in care [61,62] and increased rates of medication discontinuation [63]. These detrimental effects, therefore, necessitate improved counseling, capacity-building training for healthcare providers, and patient education to effectively manage concerns and achieve optimal care.

Individuals with AHD should be prioritized for ART initiation due to their heightened risk of mortality. It is advised that the unavailability of same-day CD4 count results should not prevent ART initiation on the same day. In our settings, four patients (2.8%) initiated ART on the same day as their diagnosis, while twenty-nine patients (20%) began treatment through rapid initiation protocols. ART should be deferred for individuals suspected of having TB or CM [7].

Prior to initiating ART, healthcare providers are advised to conduct a comprehensive discussion with patients, addressing their readiness and willingness to begin treatment, selection of the drug regimen, appropriate dosage and schedule, anticipated benefits, potential side effects, and the required follow-up and monitoring appointments. This patient-centered approach can improve treatment outcomes by enhancing adherence, promoting long-term engagement, and reducing loss to follow-up [58,64]. In our study settings, tailored counseling was provided to all individuals starting ART to promote optimal adherence.

A key concern emerging from the qualitative findings is the limited availability of medications for the management of OIs. Multiple interrelated factors appear to contribute to this challenge. First, there has been a marked reduction in the supply of OI medications from the Ethiopian Pharmaceutical Supply Agency (EPSA), the FMoH, and partner organizations supporting the national HIV program. Second, the implementation of the APTS system has restricted the practice of drug exchange between different pharmacy units within health facilities, thereby limiting access to OI medications. Third, the financial inaccessibility of these medications, which is primarily a result of patients' low socioeconomic status, further compounds the problem. These findings highlight a critical gap in the continuum of care for individuals with AHD and underscore the need for urgent targeted interventions to address this public health concern.

Another critical area is the development of differentiated service delivery models (DSD) for individuals with AHD, who require more specialized and targeted care due to their increased risk of morbidity and mortality. Similar to the approach used for key populations, care for individuals with AHD should be provided separately from the general client population. Differentiated service delivery models are considered acceptable and may contribute to improved retention rates, better viral suppression outcomes, and more efficient service delivery [65]. The implementation of differentiated antiretroviral therapy delivery models enables clinicians to dedicate more time to managing patients with AHD [26]. Furthermore, a differentiated delivery model would help improve both access to and the quality of services. Therefore, national HIV policymakers and program developers should prioritize the adaptation or development of differentiated ART delivery models tailored to the specific clinical and support needs of people with AHD.

### Clinical and public health implications

HIV program implementers and healthcare providers must consistently prioritize guideline-recommended baseline tests, including CD4 count and Cryptococcal antigen, to ensure timely AHD diagnosis and appropriate care. The initiation of ART may be accompanied by counseling once HIV diagnosis and clinical assessment are confirmed. For those suspected of

having TB or CM, the commencement of ART should be deferred. Healthcare professionals in HIV care should receive comprehensive AHD training to recognize and address patients' complex needs.

Enhanced coordination among health facilities, zonal and regional health bureaus, HIV program NGOs, the Federal Ministry of Health, and EPSA is essential to improve supply chain visibility and responsiveness. Public health authorities and policymakers are encouraged to develop and adapt differentiated service delivery models that address the diverse clinical and support needs of individuals with AHD, ensuring improved health outcomes and equitable access to comprehensive care.

### Strengths and limitations

This study employed a mixed-methods approach to facilitate a comprehensive understanding of the limited implementation of the WHO-recommended package of care for AHD. We selected a diverse group of healthcare providers from various units across the HIV care continuum to gain in-depth insights into the challenges encountered in care provision.

Despite its strengths, this study has several limitations. The lack of access to regional or national HIV case-based surveillance data, which constrained our ability to generate more robust evidence reflecting regional or national performance in the implementation of the WHO-recommended package of care for newly diagnosed PLHIV with AHD initiating ART.

### Conclusions

The implementation of the WHO-recommended package of care in the current settings remains inconsistent. Although adherence support is routinely provided to all newly diagnosed PLHIV with AHD, the delivery of other components is often impeded by various challenges. Three principal challenges have been identified: structural and organizational obstacles, service delivery constraints, and patient-related concerns as expressed by healthcare workers. The unavailability of essential diagnostic equipments, ineffective referral and linkage, interruptions in the supply of critical resources, significant shortages of OI medications, system-imposed restrictions on drug exchange practices, and the absence of specialized clinics for managing patients with AHD were critical issues. These challenges signify a serious gap in the continuum of care for individuals with AHD and underscore the urgent need for targeted interventions to address this pressing public health concern.

### Supporting information

**S1 Table. A codebook on challenges in implementing the WHO-recommended package of care for advanced HIV disease.**
(DOCX)

**S2 Table. Components of the package of care for people with advanced HIV disease.**
(DOCX)

**S1 Fig. Algorithm for providing a package of care for people with advanced HIV disease.**
(TIF)

**S1 File. Data collection tool.**
(DOCX)

### Acknowledgments

We sincerely thank the College of Health Sciences and Medicine at Dilla University for granting ethical clearance for this study. We also extend our heartfelt appreciation to the data collectors and the study participants for their cooperation and valuable time.

## Author contributions

**Conceptualization:** Temesgen Leka Lerango, Semalgn Leka Lerango, Biruk Bogale.

**Data curation:** Temesgen Leka Lerango.

**Formal analysis:** Temesgen Leka Lerango.

**Investigation:** Temesgen Leka Lerango, Semalgn Leka Lerango.

**Methodology:** Temesgen Leka Lerango, Semalgn Leka Lerango, Biruk Bogale.

**Project administration:** Temesgen Leka Lerango, Semalgn Leka Lerango, Tsion Mulat Tebeje, Daniel Sisay, Getachew Assefa Zenebe, Yohannes Addisu, Biruk Bogale.

**Software:** Temesgen Leka Lerango.

**Supervision:** Temesgen Leka Lerango, Semalgn Leka Lerango, Mesfin Abebe, Habtamu Endashaw Hareru, Yohannes Addisu, Biruk Bogale.

**Validation:** Temesgen Leka Lerango, Semalgn Leka Lerango, Mesfin Abebe, Tsion Mulat Tebeje, Habtamu Endashaw Hareru, Daniel Sisay, Getachew Assefa Zenebe, Yohannes Addisu, Biruk Bogale.

**Visualization:** Temesgen Leka Lerango, Mesfin Abebe, Tsion Mulat Tebeje, Habtamu Endashaw Hareru, Daniel Sisay, Getachew Assefa Zenebe.

**Writing – original draft:** Temesgen Leka Lerango.

**Writing – review & editing:** Temesgen Leka Lerango, Semalgn Leka Lerango, Mesfin Abebe, Tsion Mulat Tebeje, Habtamu Endashaw Hareru, Daniel Sisay, Getachew Assefa Zenebe, Yohannes Addisu, Biruk Bogale.

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
