## [Decision Letter · Decision Letter 0]

26 Dec 2025

Dear Dr. Lerango,

Thank you for submitting your manuscript to PLOS ONE. After careful consideration, we feel that it has merit but does not fully meet PLOS ONE’s publication criteria as it currently stands. Therefore, we invite you to submit a revised version of the manuscript that addresses the points raised during the review process.

We look forward to receiving your revised manuscript.

Kind regards,

Felix Bongomin, MB ChB, MSc, MMed, FECMM

Academic Editor

PLOS One

Journal Requirements:

2. You have indicated that data is available from [mohammedf@du.edu.et].  Please can we ask you to provide us with a general contact email address for the data requests, so readers can request access in perpetuity. If a general email is not available please provide a link to a website where readers can obtain access to data.

Reviewers' comments:

Reviewer's Responses to Questions

**Comments to the Author**

1. Is the manuscript technically sound, and do the data support the conclusions?

Reviewer #1: Yes

2. Has the statistical analysis been performed appropriately and rigorously?

Reviewer #1: N/A

3. Have the authors made all data underlying the findings in their manuscript fully available?

Reviewer #1: Yes

4. Is the manuscript presented in an intelligible fashion and written in standard English?

Reviewer #1: Yes

Reviewer #1: OVERALL COMMENT

This is a very useful study describing implementation gaps in the public health approach to advanced HIV disease care in Ethiopia. I have suggested a number of minor comments that would help in tightening up the paper.

One major suggestion would be for the investigators to be able to provide information on the vital status of the 145 study participants. The study was concluded over a year ago, so if possible it would be important to know if any participants had died. It would further be of interest to know timing (weeks since study end) and cause of death. But I appreciate not all of this information may be readily available.

A comment could also be made on what is generally known about HIV-associated mortality in the region/country.

SPECIFIC COMMENTS

In 2023, an estimated 39.9 million people…

- These data should be updated (2024 data are available)

“many individuals in SSA still cannot access treatment until the later stages of their illness, often presenting with advanced HIV disease”

- Data suggest that in many contexts the majority of people with AHD are not accessing treatment late, but rather are disengaging from care after having started ART. If possible, it would be helpful to describe the proportion who were newly diagnosed vs ART experienced

“International funding has been crucial in reducing new HIV infections and related deaths in SSA”

- This section would better go in the Discussion

“two-thirds (66.7%) had a CD4 count between 100 and 199. Of the 145 AHD cases, only 69 (47.6%) had a recorded baseline CD4 count”

- It needs to be clear that the 66.7% is a fraction of the 69 individuals who got a CD4.

It should also be stated how the remaining individuals were diagnosed with AHD. (In the discussion it is stated that “This implies that more than half of the cases were identified using WHO clinical staging”. Is this “implied”, or was it recorded by the investigators?)

Ref 32 is a pre-print and is now published: https://pubmed.ncbi.nlm.nih.gov/41002116/

“A key concern emerging from the qualitative findings is the limited availability of medications for the management of OIs.

- It is worth stressing that cryptococcus is particularly overlooked - none were tested for CrAG, only 33 received fluconazole primary prophylaxis.

“Median (IQR) duration from diagnosis to ART initiation was 15 days (14–18).”

- Under “Clinical and public health implications” this could be stressed as a simple area for improvement. It is stated that for some patients enrollment into ART is postponed until they come to terms with the diagnosis and are psychologically prepared to begin treatment. The presented data suggests this is the case for all patients.

- It is noted that “Prior to initiating ART, healthcare providers are advised to conduct a comprehensive discussion with patients…”. It could be emphasized that much of this counselling can be done in parallell to ART initiation, rather than sequentially. (The key issue is to rule out the risk of IRIS (screen for TB, assess for symptoms of meningitis)

Patient-related concerns are cited, but only providers were interviewed. This is noted as a limitation, but it could also be made clearer in the text (eg “Theme 3: Patient-related concerns as expressed by healthcare workers”

**Do you want your identity to be public for this peer review?** For information about this choice, including consent withdrawal, please see our Privacy Policy

Reviewer #1: No

---

## [Author Response · Author response to Decision Letter 1]

29 Dec 2025

All of our responses to the reviewer and editor comments are provided in the ‘Response to Reviewers’ file.

---

## [Editor Report · Decision Letter 1]

4 Jan 2026

Challenges in implementing the WHO-recommended package of care for advanced HIV disease in resource-constrained settings: A mixed-methods study

PONE-D-25-43416R1

Dear Dr. Lerango,

We’re pleased to inform you that your manuscript has been judged scientifically suitable for publication and will be formally accepted for publication once it meets all outstanding technical requirements.

Kind regards,

Felix Bongomin, MB ChB, MSc, MMed, FECMM

Academic Editor

PLOS One
---

## [Editor Report · Acceptance letter]

PONE-D-25-43416R1

PLOS One

Dear Dr. Lerango,

I'm pleased to inform you that your manuscript has been deemed suitable for publication in PLOS One. Congratulations! Your manuscript is now being handed over to our production team.

Kind regards,

on behalf of

Dr. Felix Bongomin

Academic Editor

PLOS One